# Research on Shape Control Characteristics of Non-oriented Silicon Steel for UCMW Cold Rolling Mill

**Hao Tao [1] , Hongbo Li [1], Jian Shao [2],\*, Jie Zhang [1], Yujin Liu [3] and Xuechang You [3]**

[1]  School of Mechanical Engineering, University of Science and Technology Beijing, Beijing 100083, China; 17888822456@163.com (H.T.); lihongbo@ustb.edu.cn (H.L.); ZhangJie@ustb.edu.cn (J.Z.)
[2]  National Engineering Research Center for Advanced Rolling Technology, University of Science and Technology Beijing, Beijing 100083, China
[3]  Shougang Zhixin Qian'an Electromagnetic Material Co., Ltd., Qian'an 064400, China; liuyujin@sgqg.com (Y.L.); youxuechang@sgqg.com (X.Y.)
\*  Correspondence: jianshao@ustb.edu.cn; Tel.: +86-10-6233-6320

**Abstract:** In order to analyze the flatness control characteristics for a certain UCMW (Universal Crown Mill with Work roll shifting) cold rolling mill, combined with the actual parameters in the field, a static simulation model of the quarter roll systems of the UCMW cold rolling mill was established by the ANSYS finite element software. The bearing roll gaps under the factors of the unit width rolling force, the roll bending force and the roll shift were calculated, which reflects the shape control characteristics and has a great influence on the friction and lubrication characteristics between the roll gaps. Additionally, the shape control strategy of the process parameters in the field was put forward. The results show that, at first, the work roll shift is the most effective shape control means, while the current-used range of the intermediate roll shift cannot make full use of the roll end contour of the intermediate roll, so the intermediate roll negative shift should be considered for shape control. At second, the excessive rolling force goes against the shape control, so the rolling force of each stand should be reasonably distributed. Finally, the shape control ability of the bending force is relatively weak, so the range of the work roll bending force should be appropriately increased.

**Keywords:** UCMW cold rolling mill; finite element simulation; roll gap friction; shape control

## 1. Introduction

In recent years, with the increasing of downstream manufacturers' requirements on the quality of cold-rolled strip products, the shape quality requirements on cold-rolled strips have become more stringent. Therefore, rolling mills with high-precision shape control ability have been more and more widely used. The UCMW (Universal Crown Mill with Work roll shifting) cold rolling mill, developed on the basis of the HC (High Crown) rolling mill, is the most representative one. It has been widely used in rolling non-oriented silicon steel with strict quality requirements because of its great performance in shape control [1–4]. The UCMW cold rolling mill is equipped with a work roll positive and negative bending roll mechanism and an intermediate roll positive bending roll mechanism to enhance the regulating ability of the bearing roll gap crown. Additionally, it has an intermediate roll axial traverse mechanism. The transverse rigidity of the bearing roll gap can be infinitely increased theoretically by adjusting the position of the intermediate roll, so the rolling stability is improved. Moreover, a single taper work roll axial traverse mechanism has been equipped to effectively control the strip edge drop and reduce the strip edge cut loss, so that the control accuracy of the strip lateral

thick difference can be improved. With reasonable cooperation of the different shape control means, it can effectively improve the shape control ability [5–10].

During the rolling process, the shape of the rolled product is actually determined by the roll gap. Therefore, the bearing roll gap is an important basis for reflecting the shape control characteristics. The shape control characteristics of the rolling mill can be fully played by researching the influence of different factors on the bearing roll gap [11–14]. For bearing roll gaps, Z. D. Han et al. [15] analyzed the influence of factors such as rolling force, bending force, and strip width on the bearing roll gaps during the strip production process by using the influence function method. X. C. Wang et al. [16] analyzed the influence of the work roll shape height on the regulation ability of the rolling mill edge drop by using the partition matrix iteration method and the influence function method. Aiming at the problem of high precision shape quality of cold-rolled strips, J. G. Cao et al. [17] proposed a control strategy which is a whole-unit integrated shape for the strip cold rolling mill.

In order to further grasp the shape control performance of the UCMW cold rolling mill during the rolling of non-oriented silicon steel, the effects of different factors on the bearing roll gap are grasped comprehensively by the finite element simulation combined with the parameters in the field, which can provide a theoretical basis for solving the shape problem in the field.

## 2. Establishment of Finite Element Model of UCMW Cold Rolling Mill Roll System

### 2.1. Rolling Mill Parameters and Model Establishment

Taking the UCMW cold rolling mill of a non-oriented silicon steel rolling production line as the research object, the static simulation model of the quarter roll system of the UCMW cold rolling mill shown in Figure 1 was established by using the general ANSYS finite element software. In the actual rolling process, the bearing roll gap of the rolling mill is influenced by many factors such as strip tension, rolling torque, material characteristics, lubrication condition, rolling parameters and rolling temperature, and these factors always change during the actual rolling process, which always influences the bearing roll gap. It is impossible for the finite element model to take into account all the factors, and the necessary assumptions and simplifications must be made:

(1) Ignore the effects of the strip tension and the rolling torque.
(2) The material of the roll is uniform and isotropic.
(3) The axis of the roll is coplanar, which is symmetrical.
(4) There is no relative sliding between the rollers.
(5) The influence of rolling temperature is not considered, namely, ignore the influence of thermal crown.
(6) Only the deformation of the roll system below the rolling line is calculated, and the interaction of the strip and the roll system is reflected by the uniformly distributed pressure.

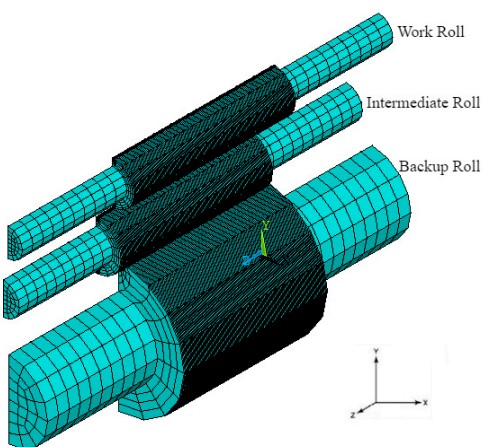

**Figure 1.** Universal Crown Mill with Work roll shifting (UCMW) roll system finite element model.

The literatures [18–20] show that this model has good applicability in calculating the influence of parameters on the roll gap during rolling. The main geometric parameters of the roll system are shown in Table 1. Among them, the shape of the backup roll is a double-sided taper roll, the shape of the intermediate roll and the work roll are single-sided taper roll, and the roll shape structure of the single-sided taper work roll is shown in Figure 2, taper length $L_1 = 150$ mm. The roll shape of the work roll and the intermediate roll are input by points, so that the actual situation of the roll shape can be accurately reflected, and the initial end roll shape parameters are shown in Table 2.

**Table 1.** The main parameters of roll system.

| Roll | Roll Neck (Diameter/mm × Length/mm) | Roll Body (Diameter/mm × Length/mm) |
| --- | --- | --- |
| Backup roll | 770 × 1090 | 1300 × 1420 |
| Intermediate roll | 320 × 975 | 490 × 1500 |
| Work roll | 280 × 1035 | 425 × 1600 |

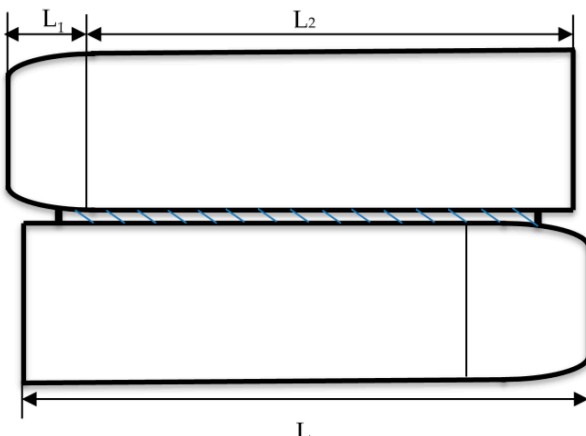

**Figure 2.** Roll shape structure of single-sided taper work roll.

**Table 2.** The initial end roll shape parameters.

| Roll Shape | Distance from the Midpoint of the Roll/mm | Radius Difference/mm |
| --- | --- | --- |
| The work roll | 800 | 0.38 |
| | 790 | 0.33102 |
| | 780 | 0.28542 |
| | 770 | 0.24320 |
| | 760 | 0.20435 |
| | 750 | 0.16889 |
| | 740 | 0.13680 |
| | 730 | 0.10809 |
| | 720 | 0.08276 |
| | 710 | 0.06080 |
| | 700 | 0.04222 |
| | 690 | 0.02702 |
| | 680 | 0.01520 |
| | 670 | 0.00676 |
| | 660 | 0.00147 |
| | 650 | 0 |
| The Intermediate roll | −750 | 2.4 |
| | −740 | 1.53473 |
| | −730 | 0.86273 |
| | −720 | 0.38326 |
| | −710 | 0.09579 |
| | −700 | 0 |

In order to make the finite element model consistent with the reality, the material characteristics of rolls use actual characteristics (Young's modulus E = 2.1 × 105 MPa, Poisson's ratio v = 0.3). The roll system is divided by the Solid45 element and the contact area between the work roll and strip is subdivided by the Solid95 element. The contact element is added to the surface between the rollers, the backup roll and work roll is Target170 element, and the intermediate roll is Contact174 element. As a result, this model is divided into 38,304 elements and 44,113 nodes.

Based on the actual force conditions of the rolling mill and the characteristics of the rolling process, these displacement constraints are imposed on the established finite element model: (1) Apply symmetry constraints and X direction displacement constraints on all nodes of the YZ plane of the roll system: $UX = 0$; (2) Apply a Y direction displacement constraint at the midpoint of the top contact line on the backup roll: $UY = 0$; (3) Apply a Z direction displacement constraint at the geometric centers of the work roll, intermediate roll and backup roll: $UZ = 0$. The loads are further applied to the model. (1) Rolling force: it is applied on the upper contact line of the work roll as a uniform load. (2) Bending force of the intermediate roll and the work roll: it is applied in the center of the journal section of both ends of the intermediate roll and the work roll as a concentrated force. The bearing condition of the roller system and the bearing roll gap are shown in Figure 3. The deformation of the work roll surface is calculated by finite element simulation, and then the bearing roll gap can be calculated.

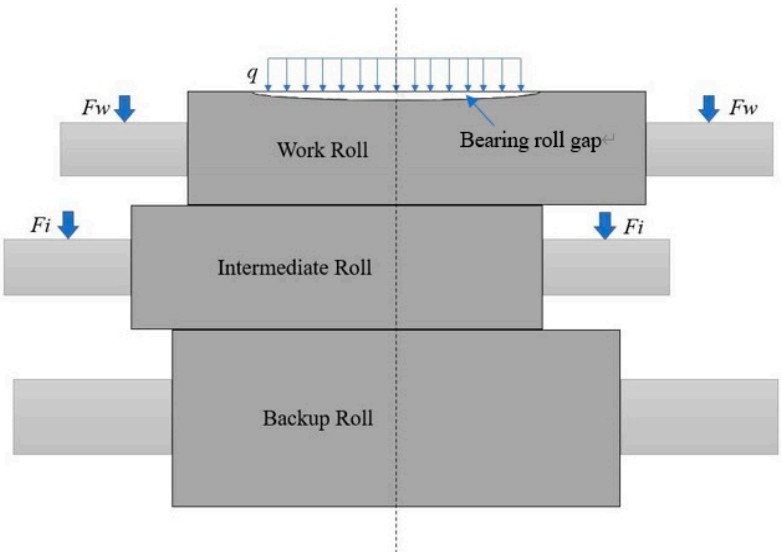

**Figure 3.** The bearing condition of the roller system.

## 2.2. Field Data Acquisition and Simulation Condition Design

In order to analyze the shape control characteristics of the UCMW mill, this paper mainly focuses on several main means of shape control, such as: unit width rolling force $q$, intermediate roll bending force $Fi$, work roll bending force $Fw$, intermediate roll shift $Si$ and work roll shift $Sw$. Among them, the definition of the intermediate roll shift $Si$ and the work roll shift $Sw$ is shown in Figure 4. According to the date tracked and the various parameters of each stand in the field, the simulation condition is designed as shown in Table 3. Other factors are controlled as the initial state when researching the influence of a single factor, setting $Fi = 80$ kN, $Fw = 80$ kN, $Si = 0$ mm, $Sw = 0$ mm, $q = 9$ kN/mm as the initial state.

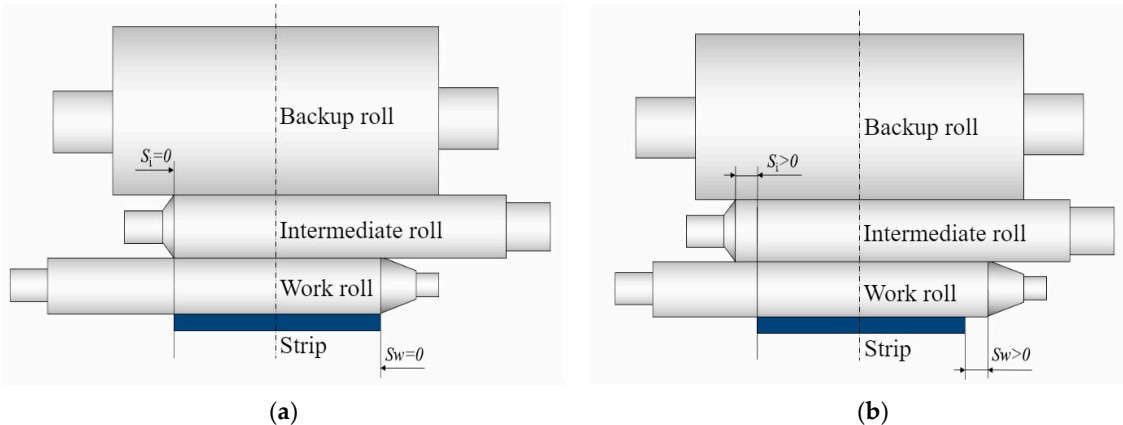

**Figure 4.** Roll shift amount. (**a**) Zero shifting roll position, (**b**) positive shifting roll position.

**Table 3.** Simulation condition.

| Parameters | On-Site Value Range | Simulation Value |
|---|---|---|
| Strip width B/mm | 1000~1200 | 1000, 1100, 1200 |
| Unit width rolling force $q$/(kN/mm) | - | 6, 9, 12 |
| Intermediate roll bending force $Fi$/kN | 80~220 | 80, 150, 220 |
| Work roll bending force $Fw$/kN | 80~220 | 80, 150, 220 |
| Intermediate roll shift $Si$/mm | 0~80 | −40, 0, 40, 80 |
| Work roll shift $Sw$/mm | −80~80 | −80, −60, −40, 0, 40, 80 |

## 3. Experimental Verification of the Model

In order to verify the accuracy of the model, the lateral thick difference of the roll gap calculated by the model is compared with the lateral thick difference of the strip obtained by field measurement. It is found that the model calculation result is in the same trend as the field measurement result, and the error is also acceptable. One of the working condition parameters is that the strip width B = 1240 mm, the rolling force Q = 11,384.7 kN, the work roll bending force $Fw$ = 195 kN, the intermediate roll bending force $Fi$ = 228 kN, the work roll shift $Sw$ = −70 mm and the intermediate roll shift $Si$ = 10 mm. The calculation conditions of the simulation are basically identical with the rolling conditions of the first stand of the non-oriented silicon steel grade 50SW1300 in the field. The comparison between the simulation results and the measured results is shown in Figure 5.

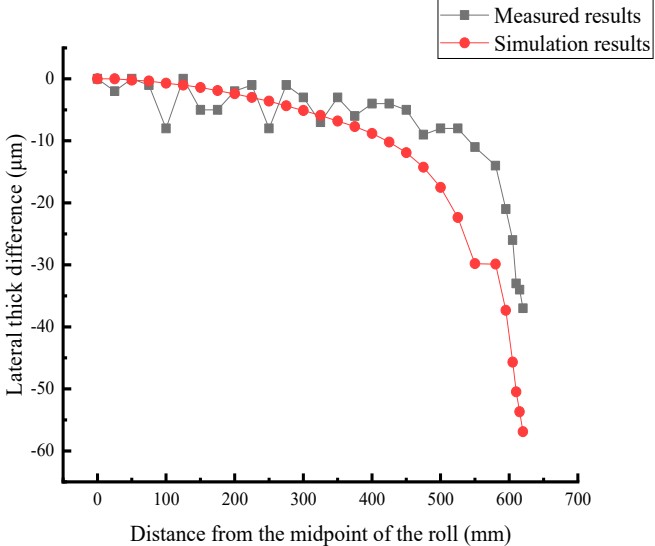

**Figure 5.** The comparison between the simulation results and the measured results.

It can be seen that although the simulation results are larger than the measured results, whether the crown or the edge drop, the error is also acceptable. The simulation result is larger because this model does not consider some factors of the strip (strip deformation resistance, tension, etc.), so the simulation result is satisfactory. This model is suitable for analyzing the influence of the parameters on the deformation of the roll system.

## 4. Analysis of Shape Control Characteristics for UCMW Cold Rolling Mill

### 4.1. Effect of Unit Width Rolling Force on Roll Gap Shape

The effect of the unit width rolling force on roll gap shape is shown in Figure 6. The results of the 1100 mm width strip in Figure 6a show that the bearing roll gap crown decreases from 207.8 μm to 74.6 μm (i.e., 64.1% drop) and the bearing roll gap edge drop decreases from 96.4 μm to 39.7 μm (i.e., 58.8% drop) when the unit width rolling force decreases from 12 kN/mm to 6 kN/mm, which easily causes the shape problem of excessive crown, especially the excessive edge drops for the S1 and S2 stands with the relatively large rolling forces. Then, it can be seen from Figure 6b that at the same unit width rolling force, the roll gap crown changes little with the increase in strip width. As shown in Figure 6c, the variation of the roll gap crown corresponding to the variation of the unit width rolling force is basically the same for the strip with different widths, which indicates that there is little difference between the crown and edge drop control when using the same unit width rolling force to roll the non-oriented silicon steel with different widths.

### 4.2. Effect of Intermediate Roll Bending Force on Roll Gap Shape

The effect of intermediate roll bending force on roll gap shape is shown in Figure 7. The analysis results of the 1100 mm width strip in Figure 7a show that the bearing roll gap crown decreases from 141.3 μm to 135.5 μm (i.e., 4.1% drop) and the bearing roll gap edge drop decreases from 68.1 μm to 66.9 μm (i.e., 1.8% drop) when the intermediate roll bending force increases from 80 kN to 220 kN, which is the minimum and maximum values of the actual rolling parameters. As shown in Figure 7b, the variation of the roll gap crown corresponding to the variation of the intermediate roll bending force is basically the same for the strip with different widths, which indicates that the intermediate roll bending force has basically no effect on the shape control within the current-used range of the intermediate roll bending force in the field.

### 4.3. Effect of Work Roll Bending Force on Roll Gap Shape

The effect of the work roll bending force on roll gap shape is shown in Figure 8. The results of the 1100 mm width strip in Figure 8a show that the bearing roll gap crown decreases from 141.3 μm to 80 μm (i.e., 43.4% drop) and the bearing roll gap edge drop decreases from 68.1 μm to 49.1 μm (i.e., 27.9% drop) when the work roll bending force increases from 80 kN to 220 kN, which are the minimum and maximum values of the actual rolling parameters. It indicates that the shape control ability of the UCMW rolling mill is still not strong within the current-used range of the roll bending force in the field, but the shape control ability of the work roll bending force is more than 10 times that of the intermediate roll bending force within the same range of roll bending force. As shown in Figure 8b, the variation of the roll gap crown corresponding to the variation of the work roll bending force is basically the same for the strip with different widths.

It can be seen that the work roll bending force should be the main control means based on the results of the effects of the intermediate roll bending force and the work roll bending force on the roll gap shape. More importantly, the current-used range of the work roll bending force in the field should be increased to better exert the shape control characteristics of the bending force.

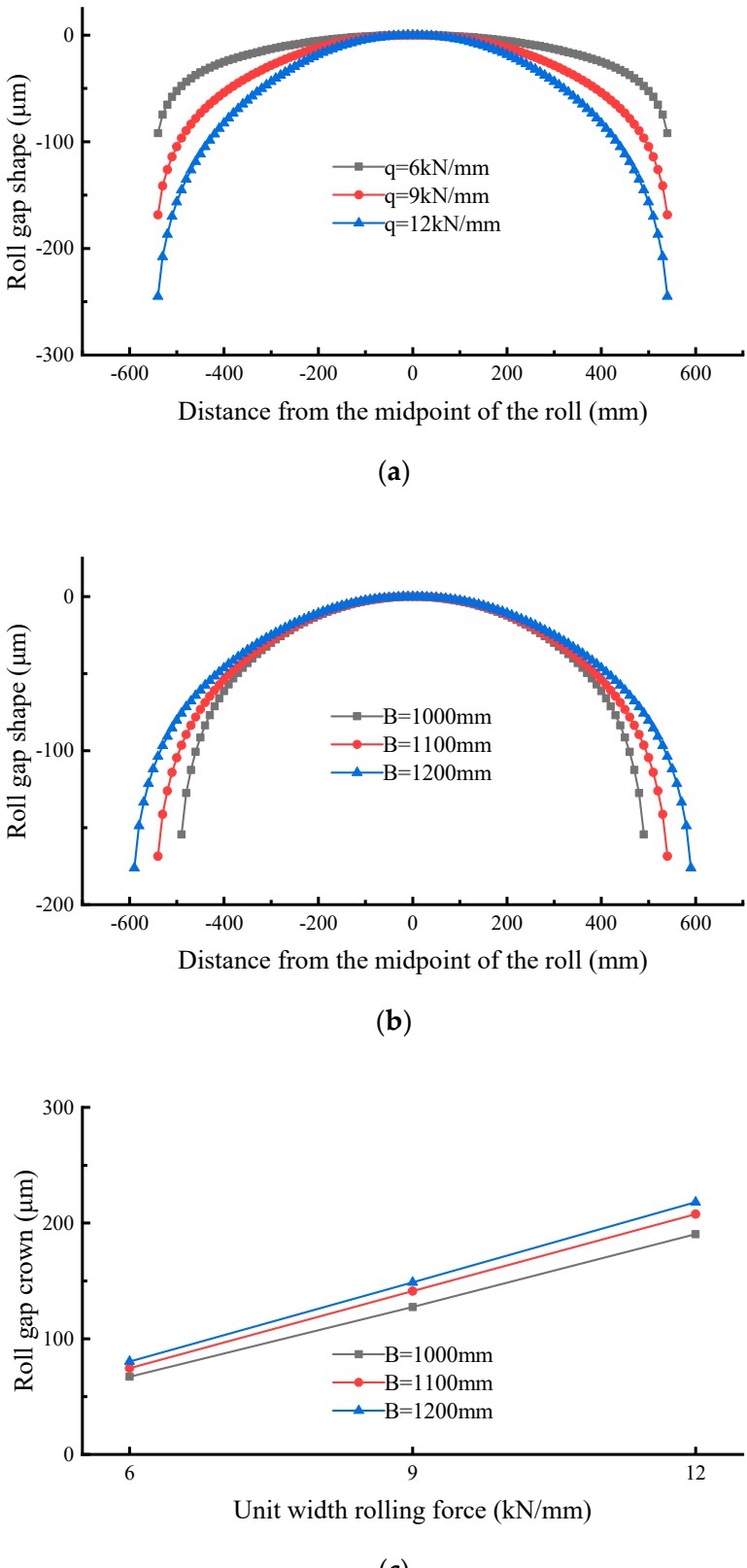

**Figure 6.** Effect of unit width rolling force on roll gap shape. (**a**) Roll gap shape under different unit width rolling forces, (**b**) roll gap shape of strip with different widths under the same unit width rolling force, (**c**) effect of unit width rolling force on roll gap crown.

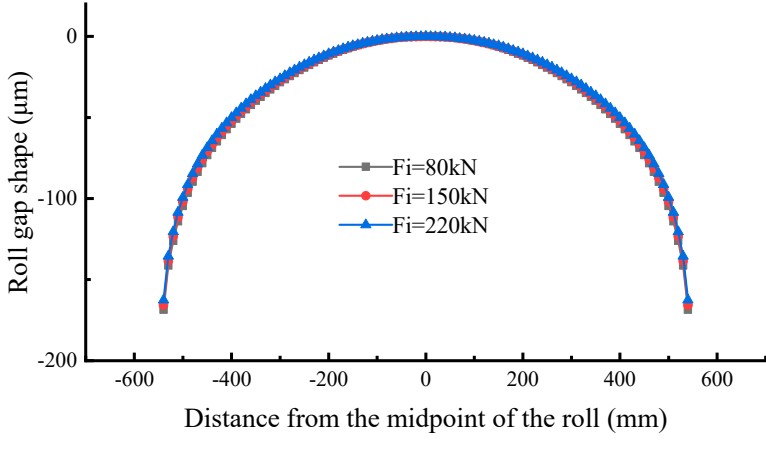

(**a**)

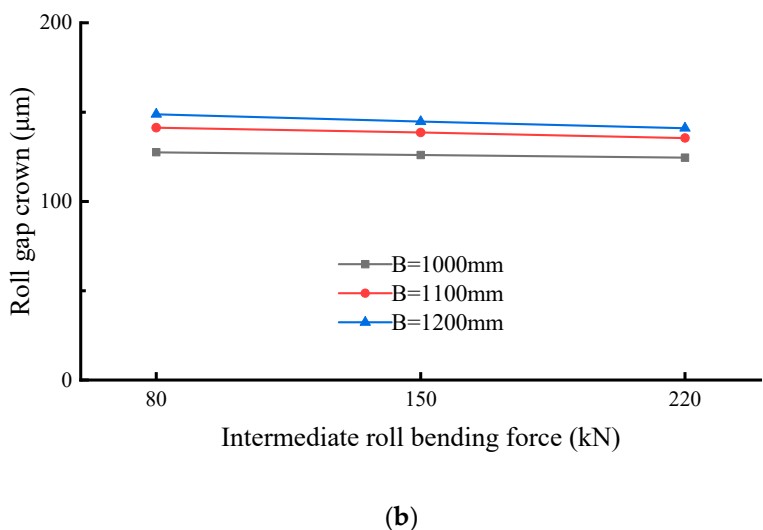

(**b**)

**Figure 7.** Effect of intermediate roll bending force on roll gap shape. (**a**) Roll gap shape under different intermediate roll bending forces, (**b**) effect of intermediate roll bending force on roll gap crown.

*4.4. Effect of Intermediate Roll Shift on Roll Gap Shape*

The effect of the work roll bending force on roll gap shape is shown in Figure 9. The results of the 1100 mm width strip in Figure 9a show that the bearing roll gap crown decreases from 230.2 μm to 141.3 μm (i.e., 38.6% drop) and the bearing roll gap edge drop decreases from 90.8 μm to 68.1 μm (i.e., 25% drop) when the intermediate roll shift decreases from 80 mm to 0 mm, which is the current-used range of intermediate roll shift in the field. As shown in Figure 9b, the variation of the roll gap crown corresponding to the variation of the intermediate roll shift is slightly different for the strip with different widths; the variation of the 1000 mm width strip is 86.2% of the 1200 mm width strip. The above results show that within the current-used range of the intermediate roll shift in the field, the smaller the roll shift is, the better the shape control ability shows. In fact, the current-used range of the intermediate roll shift cannot make full use of the roll end contour of the intermediate roll. As shown in Figure 9a, the roll gap crown and edge drop are greatly decreased when the intermediate roll is negatively shifted. They decrease by 41.5% and 21.1%, respectively, when the intermediate roll shift decreases from 0mm to −40 mm, which indicates that the intermediate roll negative shift should be considered to make full use of the roll end contour of the intermediate roll to shape control.

*4.5. Effect of Work Roll Shift on Roll Gap Shape*

The effect of the work roll bending force on roll gap shape is shown in Figure 10. The results of the 1100 mm width strip in Figure 10a show that the work roll positive shift (i.e., $Sw > 0$ mm) has little influence on the roll gap crown and edge drop. The decreased ratio of the crown and edge drop between $Sw = 80$ mm and $Sw = 0$ mm is 20.6% and 12.9%. However, the roll gap edge region has a significant change which makes the roll gap crown and edge drop significantly decreased, especially the edge drop, as the work roll negative shift (i.e., $Sw < 0$ mm) increases. The roll gap crown and edge drop decrease by 22.8% and 20.2%, respectively, when the work roll shift decreases from 0 mm to −40 mm, which indicates that the control ability of the work roll shift to the roll gap crown is smaller than the intermediate roll shift, but it is stronger for roll gap edge drop compared with the results of the intermediate roll shift $Si = −40$ mm. The roll gap edge region has a significant local change when the work roll shift is $Sw = −60$ mm and $Sw = −80$ mm. The edge drop value at $Sw = −60$ mm is 44.8% lower than $Sw = 0$ mm. Moreover, the phenomenon of the local increase in the roll gap edge appears when the work roll shift is $Sw = −80$ mm, which indicates that the more the roll end contour enters in the strip width range, the stronger the local control ability of the strip edge drop shows.

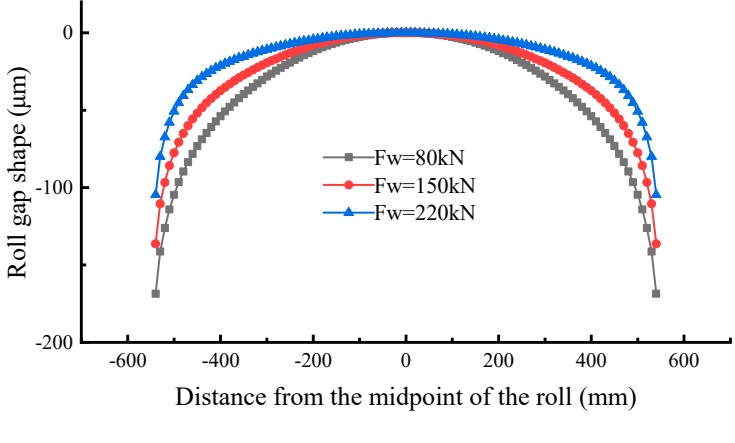

(**a**)

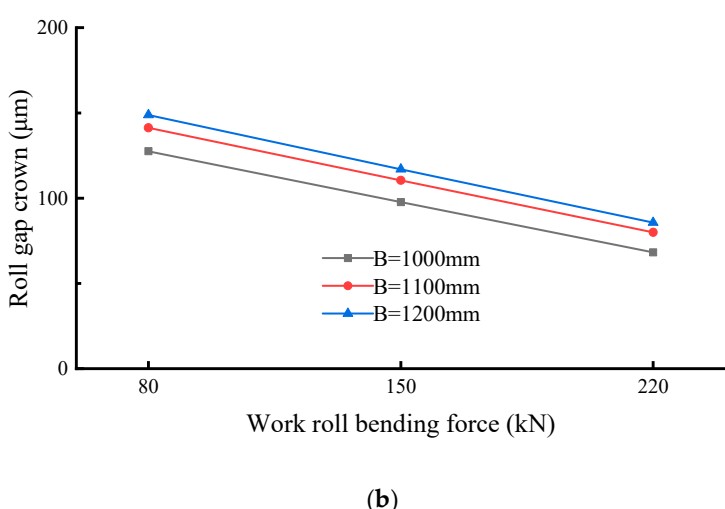

(**b**)

**Figure 8.** Effect of work roll bending force on roll gap shape. (**a**) Roll gap shape under different intermediate roll bending forces, (**b**) effect of intermediate roll bending force on roll gap crown.

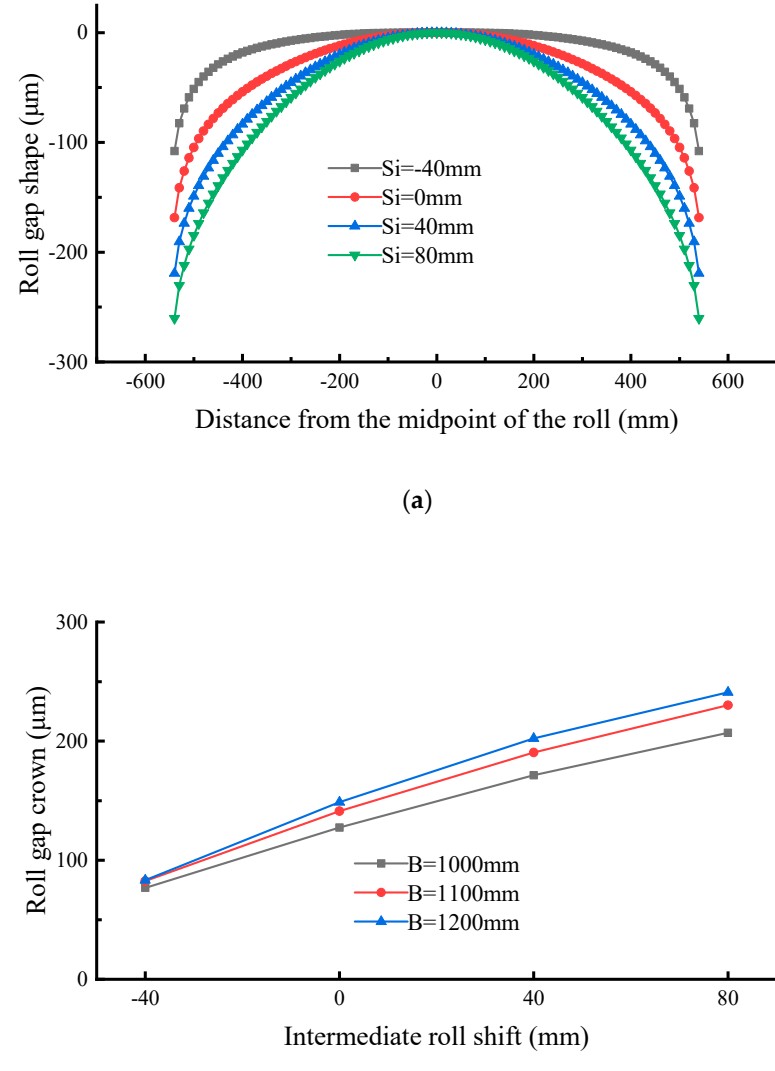

(**a**)

(**b**)

**Figure 9.** Effect of intermediate roll shift on roll gap shape. (**a**) Roll gap shape under different intermediate roll shifts, (**b**) effect of intermediate roll shift on roll gap crown.

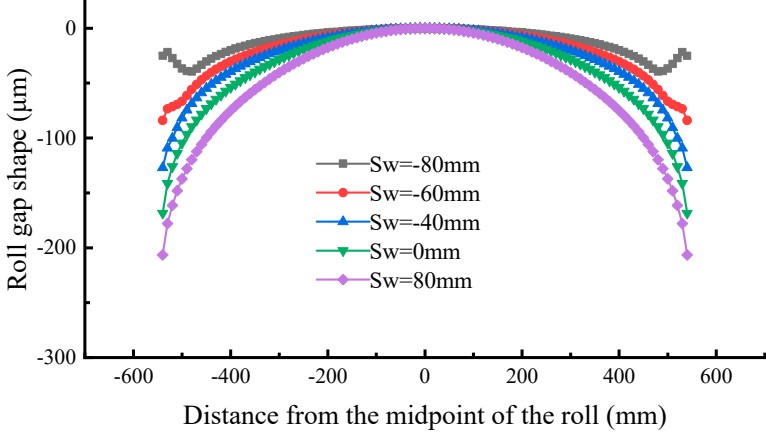

(**a**)

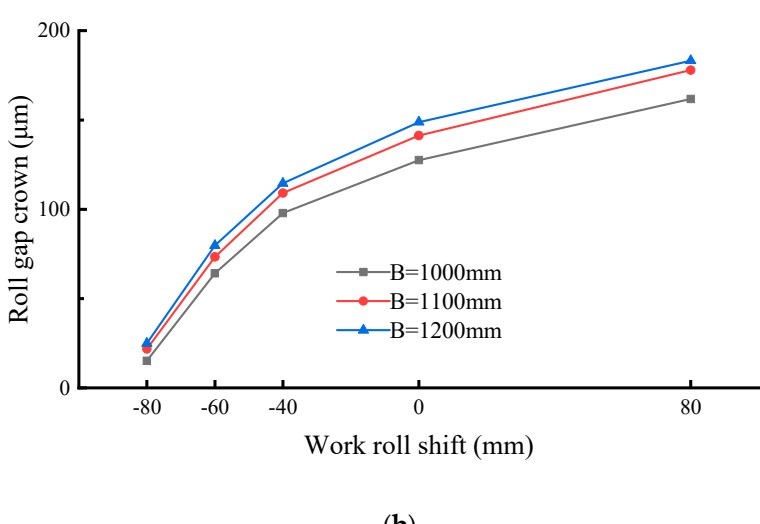

(**b**)

**Figure 10.** Effect of work roll shift on roll gap shape. (**a**) Roll gap shape under different work roll shifts, (**b**) effect of work roll shift on roll gap crown.

## 5. Conclusions

Through the above finite element simulation analysis, the conclusions are:

(1) According to the results of the effect of the unit width rolling force on roll gap shape, it can be seen that the rolling force of the S1 and S2 stands is relatively large in the field, which easily causes the shape problem of excessive crown and edge drop. Meanwhile, the friction condition between the roll and the strip will deteriorate because of the large rolling force. Therefore, the rolling force of each stand needs to be distributed appropriately, with an aim to avoid too much rolling force in a certain stand.

(2) The results of the intermediate roll bending force and the work roll bending force show that the intermediate roll bending force has basically no effect on the shape control. Although the shape control ability of the work roll bending force is more than 10 times that of the intermediate roll bending force, it is still insufficient. Therefore, the range of the work roll bending force should be appropriately increased to fully use the control ability of the bending force.

(3) The smaller the intermediate roll shift is, the better the control ability of the crown and edge drop shows within the current-used range of the intermediate roll shift. Moreover, the means of the

intermediate roll negative shift should be taken to make use of the roll end contour of the intermediate roll for shape control.

(4) The influence of the work roll positive shift on the roll gap crown and edge drop is small. However, the work roll negative shift has significant control on the crown and edge drop because the roll end contour can enter in the strip width range. Notably, the more the roll end contour enters the strip width range, the stronger the local control ability of the strip edge drop shows.

**Author Contributions:** Conceptualization, H.L., J.S. and J.Z.; Date curating, H.T.; investigation, H.T.; methodology, H.T.; resources, Y.L. and X.Y.; writing—original draft preparation, H.T.; writing—review and editing, H.L., J.S. and J.Z.; project administration, H.L. and J.Z. All authors have read and agreed to the published version of the manuscript.

**Funding:** This research was funded by the National Key Technology R&D Program of the 12th Five-year Plan of China, grant number 2015BAF30B01. The APC was funded by University of Science and Technology Beijing.

**Acknowledgments:** First of all, I would like to thank the laboratory teachers and brothers for their help in my study, and they have given me great help in simulation processing. Secondly, thank the Shougang Zhixin Qian'an Electromagnetic Material Co., Ltd. for providing us with cold-rolled non-oriented silicon steel for data testing. Finally, I am very grateful to the three reviewers for their valuable comments, which made this article a great improvement in details.

**Conflicts of Interest:** The authors declare no conflict of interest.

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
