# Peer review of "Research on Shape Control Characteristics of Non-oriented Silicon Steel for UCMW Cold Rolling Mill"

_metals, doi:10.3390/met10081066_

Round 1

Reviewer 1 Report

The manuscript was well revised by authors from last submitting to this system. The authors have responded to all my comments. This manuscript lacks of scientific novelty. However, this manuscript can be helpful for understanding the shape control performance of the work rolls and has certain significance for technologies of cold rolling of the non-oriented silicon steels.

Reviewer 2 Report

Dear authors.
All my previous comments were considered in the revised manuscript.
The manuscript could be accepted for the publication.

This manuscript is a resubmission of an earlier submission. The following is a list of the peer review reports and author responses from that submission.

Round 1

Reviewer 1 Report

The authors present analyses of a UCMW Cold rolling mill. Unfortunately, this paper is quite difficult to understand if you are not familiar with this particular design process. I am also unsure of what scientific value this article brings. The motivation and the discussed problem need to be presented much better, in order for the reader to understand the different parameters discussed. Unfortunately, the reviewer cannot recommend publication.

Reviewer 2 Report

  1. Boundary conditions of FEM simulation are not correct. Line 106: “Rolling force: It applies on the symmetry plane of the roll system as a uniform load”. In the real process rolling force does not act in the symmetry plane of the roll system. Rolling force acts inside of the plastic deformation zone at the contact of strip with work roll. Load is usually non-uniform. All this greatly affects the shape of the roll gap. The authors need to first calculate the plastic deformation zone for different width of strips, and then interpolate the resulting loads on the roll system to calculate the shape of the roll gap for each case.
  1. In the real process the work rolls are not cylindrical because of special CVC (Continuously Variable Crown) profile. Authors need to perform calculations with taking into account real rolls profiles.
  2. Material characteristics of rolls (Young's modulus, Poisson's ratio) should be presented.
  3. Roll gap shape, which is changed in microns, is very sensitive to sizes of finite elements during simulations. It is necessary to provide information on the dimensions of the finite elements, especially in places of application of the load.
  4. The vertical axis in the Figure 2 (a and b) should have units.
  5. Experimental results should be presented.
  6. The scientific novelty of the article should be formulated.

Reviewer 3 Report

Dear authors,

I sure that your manuscript with the title “Analysis of Shape Control Characteristics for UCMW Cold Rolling Mill Non-Oriented Silicon Steel” will be interesting for both academic an industrial community. The problem related ship with cold rolling of high quality grades of NO steel is very actual nowadays. But, it seems to me that results obtained only on the base the computer simulation without the back up experimental data will be not trustworthy.

  • What is mean UCMW or HC rolling mill? All abbreviation must be definition minimum one time in the text.
  • Please define each roll In Fig. 1 (Backup, intermediate and work roll).
  • Which information was shown in Fig. 2. What is mean “Roll shape” and which units description this parameter? Why “Roll shape” in Fig. 2a and 2b is visual different? Maybe it would be better if “Roll shape” define as distance from roll neck to surface roll body.
  • Please show in the Fig.3 where is the cold rolled steel sheet between the roll?
  • Which roll speed was used in your simulation?
  • Why mechanical properties of real steels was not used in the simulation. All computer simulation in the ANSYS need the physical parameters of materials which are used in simulating process.
  • Please show the roll gap shape on the modelling UCMW cold rolling mill.
  • The result in Fig. 4 – Fig. 8 is not understandable without some visualisation of modelling parameters on (or between) the rolls.
  • Please explain why you submitted your manuscript to the journal “Metals” if your results was not back up by experimental data? Maybe this scientific work will be better submit to the journal which focused only the computer simulation?
  • Haw the calculation results obtained in Fig. 4 – Fig. 8 will be depend on the mechanical characteristic of different grade of Non-oriented steel? Different steels contain different amount of silicon which very influenced their hardness, yield stress and tensile stress.
  •